# Asymmetric recruitment and actin-dependent cortical flows drive the neuroblast polarity cycle

**Chet Huan Oon, Kenneth E Prehoda***

Department of Chemistry and Biochemistry, Institute of Molecular Biology, University of Oregon, Eugene, United States

**Abstract** During the asymmetric divisions of *Drosophila* neuroblasts, the Par polarity complex cycles between the cytoplasm and an apical cortical domain that restricts differentiation factors to the basal cortex. We used rapid imaging of the full cell volume to uncover the dynamic steps that underlie transitions between neuroblast polarity states. Initially, the Par proteins aPKC and Bazooka form discrete foci at the apical cortex. Foci grow into patches that together comprise a discontinuous, unorganized structure. Coordinated cortical flows that begin near metaphase and are dependent on the actin cytoskeleton rapidly transform the patches into a highly organized apical cap. At anaphase onset, the cap disassembles as the cortical flow reverses direction toward the emerging cleavage furrow. Following division, cortical patches dissipate into the cytoplasm allowing the neuroblast polarity cycle to begin again. Our work demonstrates how neuroblasts use asymmetric recruitment and cortical flows to dynamically polarize during asymmetric division cycles.

DOI: https://doi.org/10.7554/eLife.45815.001

*For correspondence:
prehoda@uoregon.edu

**Competing interests:** The authors declare that no competing interests exist.

## Introduction

*Drosophila* neuroblasts dynamically polarize to segregate fate determinants while dividing asymmetrically (*Homem and Knoblich, 2012*; *Knoblich, 2010*; *Prehoda, 2009*; *Venkei and Yamashita, 2018*). Cortical polarization during mitosis allows fate determinant containing cortical domains to be separated by the cleavage furrow during cytokinesis. Following division, fate determinant segregation causes one daughter cell to retain the neuroblast fate and to undergo further asymmetric divisions, while the other takes on a differentiated fate to populate the central nervous system. The catalytic activity of atypical Protein Kinase C (aPKC), a component of the animal cell polarity Par complex, is central to this process, and must be localized to the neuroblast's apical cortex during mitosis. Phosphorylation of neuronal fate determinants displaces them from the membrane, ensuring that they are restricted to the basal cortex to be segregated into the differentiating daughter cell (*Atwood and Prehoda, 2009*; *Bailey and Prehoda, 2015*; *Betschinger et al., 2003*; *Lang and Munro, 2017*; *Rolls et al., 2003*). During each asymmetric neuroblast division, aPKC cycles between polarized and unpolarized states. Here we examine the dynamic processes that underlie aPKC polarization and depolarization during neuroblast asymmetric division cycles.

Neuroblasts begin asymmetric division with aPKC in the cytoplasm (*Hannaford et al., 2018*). By metaphase, aPKC accumulates at a cortical domain around the apical pole where it directs the polarization of differentiation factors such as Miranda and Numb to the basal cortex (*Homem and Knoblich, 2012*; *Knoblich, 2010*; *Prehoda, 2009*). Preferential targeting of aPKC to the apical cortex could explain neuroblast polarization, although little is known about how this process might occur. Furthermore, asymmetric targeting as a polarization mechanism contrasts with the dynamics of aPKC polarization in the early worm embryo in which aPKC is localized to both the anterior and

posterior domains of the worm cortex before sperm entry (*Lang and Munro, 2017*; *Tabuse et al., 1998*; *Wang et al., 2017*). Directional transport from the posterior to anterior cortical domain (i.e. cortical flow), potentially through the activity of actomyosin (*Munro et al., 2004*), is thought to play a key role in the worm embryo. It has been unknown whether cortical flows play any role in aPKC polarization in neuroblasts. Furthermore, the nature of the aPKC cortical recruitment process has not been described.

Because neuroblasts repeatedly cycle between polarized (apical aPKC at metaphase) and unpolarized (interphase cytoplasmic aPKC) states, depolarization is a necessary step in the neuroblast

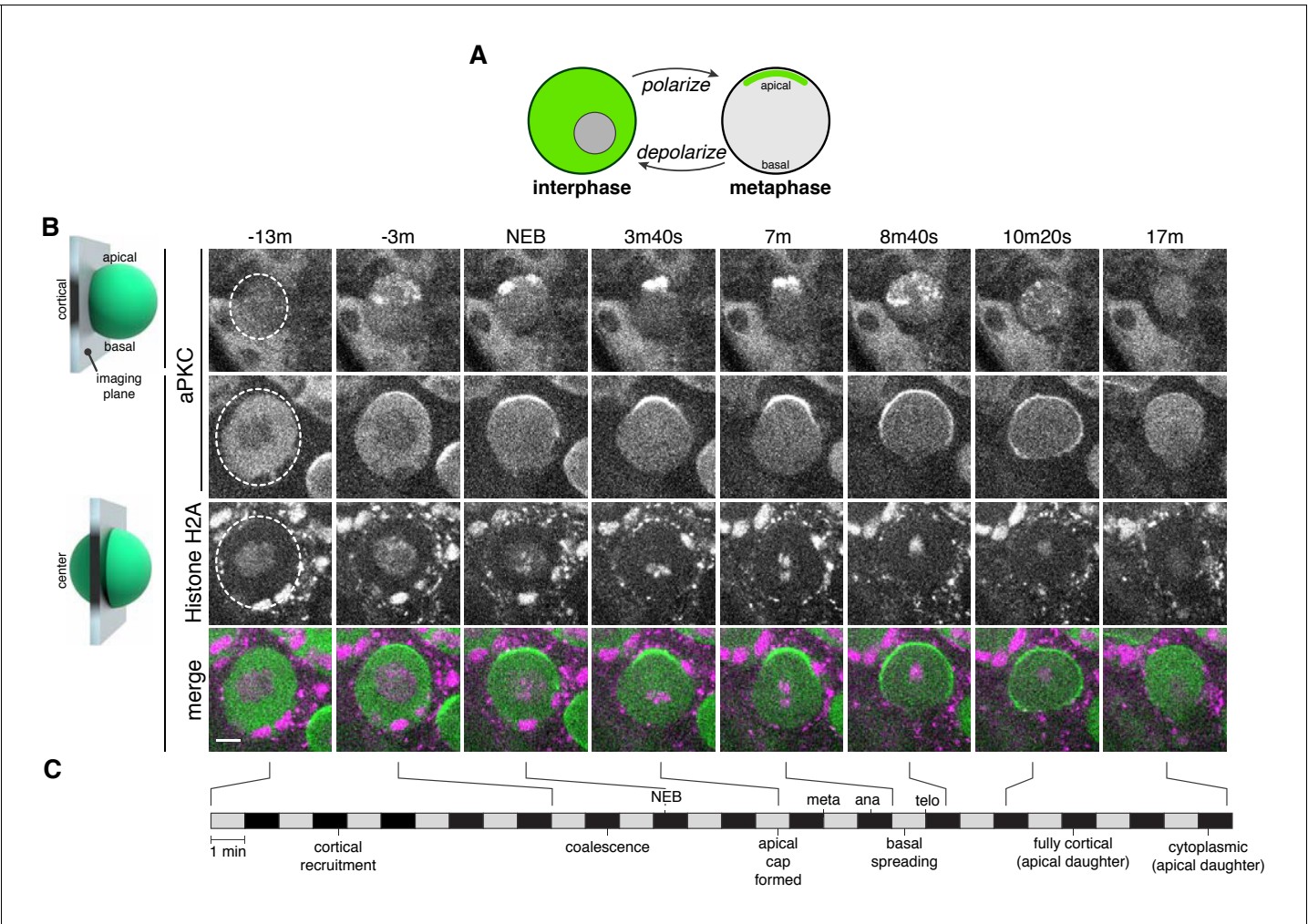

**Figure 1.** The neuroblast polarity cycle is a dynamic, multistep process. (A) Schematic of the neuroblast polarity cycle. Neuroblasts transition between unpolarized, cytoplasmic aPKC in interphase, to an apical cortical domain tightly focused around the apical pole in metaphase, the canonical neuroblast polarity state, during repeated asymmetric divisions. (B) Frames from *Figure 1—video 1* showing 1.5 μm maximum intensity projections of aPKC-GFP signal along the cortical edge ('cortical'; top row) and center ('center'; rows 2–4) of a neuroblast. A maximum intensity project of RFP-Histone H2A signal through the center of the cell, along with a merge of GFP and RFP central projections, are also shown. The outline of the neuroblast is highlighted with a dashed circle in the first column. Time is shown relative to nuclear envelope breakdown. (C) Timeline of the neuroblast polarity cycle with cell cycle hallmarks (NEB, nuclear envelope breakdown; meta, metaphase; ana, anaphase; telo, telophase) marked above the timeline and polarization events below.

DOI: https://doi.org/10.7554/eLife.45815.002

The following video and figure supplement are available for figure 1:

**Figure supplement 1.** Cortical localization in fixed neuroblasts.

DOI: https://doi.org/10.7554/eLife.45815.003

**Figure 1—video 1.** Localization dynamics of the Par complex component aPKC during neuroblast asymmetric division.

DOI: https://doi.org/10.7554/eLife.45815.004

polarity cycle (*Figure 1A*). However, little is known about the events that follow metaphase that regenerate the unpolarized state. These events may be especially important for asymmetric division because the localization of aPKC at metaphase is distant from the site of cleavage furrow formation in anaphase, the exclusion point for basal fate determinants. Understanding how metaphase polarity is disassembled may provide insight into the mechanism by which determinants are prevented from occupying the apical cortex that becomes the self-renewed neuroblast following division.

We have investigated how neuroblasts transition between polarity states – the neuroblast polarity cycle – to gain insight into the mechanisms by which metaphase polarity is formed and disassembled. We have sought to determine whether neuroblast polarity results from direct recruitment from the cytoplasm, or if the process requires additional steps. Likewise, does depolarization occur simply from direct exchange from the apical cortex into the cytoplasm? Furthermore, we have examined the role of the actin cytoskeleton in neuroblast polarization and depolarization. The dynamic steps in neuroblast polarization that we have discovered provide further insight into the mechanisms underlying animal cell polarity and a new framework for using the neuroblast as a polarity model system.

## Results

### The neuroblast polarity cycle is a dynamic, multistep process

We investigated the divisions of neuroblasts from *Drosophila* larval brain lobes (*Homem and Knoblich, 2012*), first focusing on a GFP fusion of aPKC (aPKC-GFP) (*Besson et al., 2015*), as its catalytic activity is the direct output of the Par complex (*Atwood and Prehoda, 2009*; *Bailey and Prehoda, 2015*). We simultaneously imaged an RFP fusion of Histone H2A (RFP-H2A) to assess the cell cycle stage. To identify as much of the dynamics of the neuroblast polarization process as possible, we imaged the process every 20 s or faster, the maximum acquisition frequency that yielded sufficient signal and little photobleaching. Furthermore, we collected optical sections throughout the full volume of the cell to visualize sections in the center along with those at the cortical edge and to allow for full three-dimensional projections at each time point. These data reveal a highly dynamic process that begins with aPKC in the cytoplasm as cells entered mitosis (*Figure 1B and C*; *Figure 1—video 1*). Near the time when chromosome condensation became apparent, discrete aPKC foci appeared on the cortex, preferentially in apical hemisphere. We also observed aPKC foci in three dimensional projections of fixed, wild type prophase neuroblasts using an anti-aPKC antibody (*Figure 1—figure supplement 1*). Near metaphase, the aPKC cortical foci, which by then had grown into larger patches, moved toward the apical pole in a concerted fashion, coalescing into an 'apical cap', the metaphase neuroblast polarity state. The aPKC apical cap remained until shortly after anaphase onset at which point the cap disassembled by rapid spreading of cortical patches toward the contracting cleavage furrow. No aPKC signal was detected on the cortex of the basal ganglion mother cell (*Figure 1B*; *Figure 1—video 1*). In addition, the cortical aPKC in the newborn neuroblast daughter rapidly dissipated into the cytoplasm at the end of mitosis. The overall polarity cycle, from the initial appearance of cortical foci to dissipation occurred in 28.8 ± 8.2 min (n = 20 neuroblasts from four larvae).

These data reveal previously unrecognized complexity in neuroblast polarization and depolarization processes. We speculate that previous studies failed to observe these dynamics because of their transient nature, and furthermore, the discontinuous nature of the cortical aPKC signal is less visible in central optical sections compared to those along the cortical edge (*Figure 1B*; *Figure 1—video 1*). In the following sections we examine the neuroblast polarity cycle in more detail.

### Asymmetric cortical recruitment yields a discontinuous, unorganized structure

High frame rate projections of the full neuroblast volume revealed that the initial step in aPKC polarization is the formation of discontinuous patches on the apical cortex (*Figure 1B*; *Figure 1—video 1*; *Figure 2A*). Apical targeting begins in early prophase and ends shortly before nuclear envelope breakdown (NEB; as assessed by the appearance of aPKC-GFP in the nucleus) with an overall time of 11.1 ± 6.2 min (n = 20). We observed the first small cortical foci in early prophase when chromosome condensation became apparent. Focus formation was heavily biased toward the apical cortical

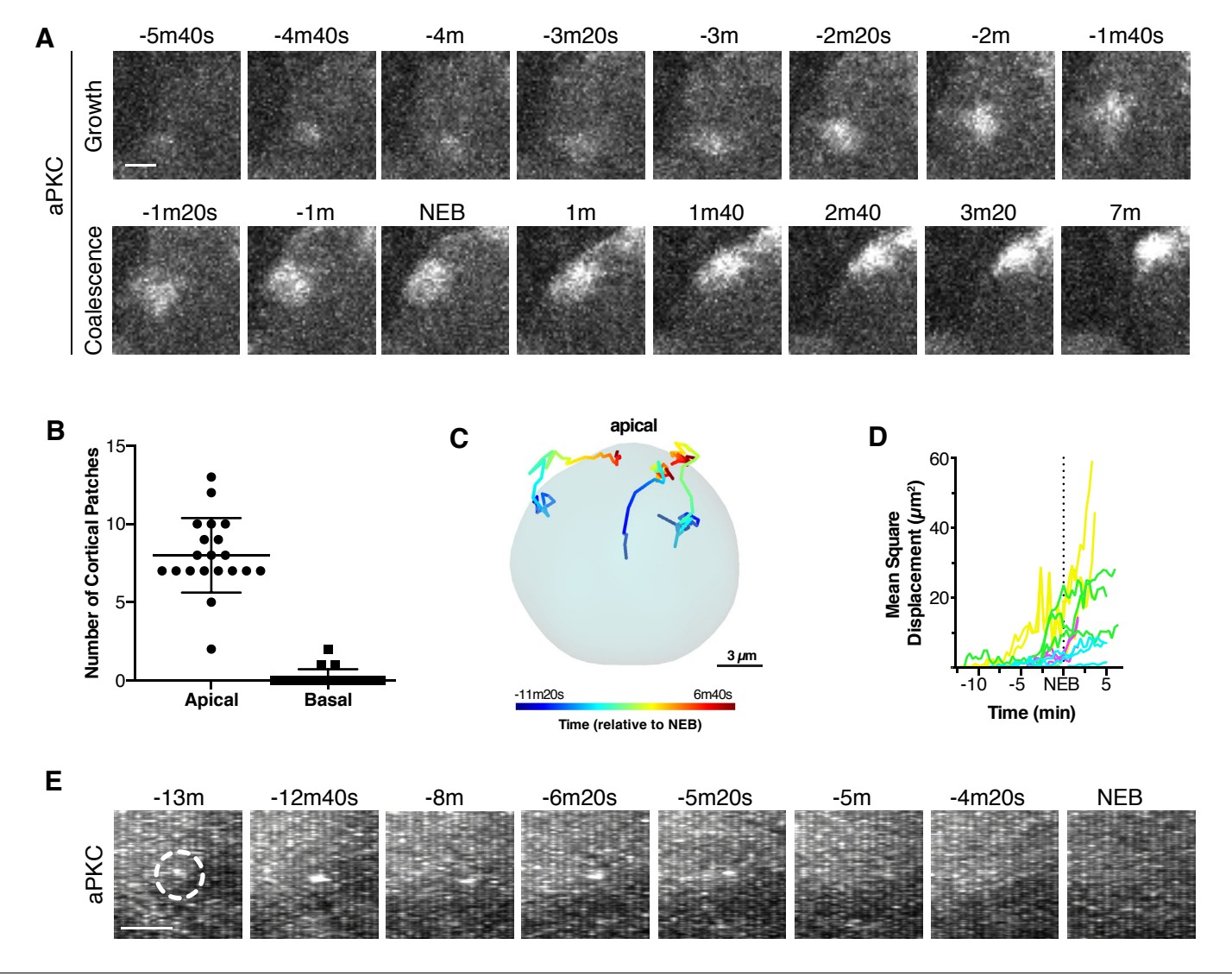

**Figure 2.** Apically directed cortical recruitment and patch coalescence. (A) Example of aPKC-GFP cortical patches during growth and coalescence phases. Scale bar 2 μm. (B) The number of aPKC-GFP cortical patches in the apical and basal hemispheres immediately before cortical flow begins. Each point represents a distinct neuroblast (taken from four larvae). Bars represent one standard deviation from the mean. Data are included in *Figure 2—source data 1*. (C) Example patch trajectories during coalescence from particle tracking. Cell outline is shown in light blue. (D) Mean square displacement of several different patches identified by particle tracking as a function of time. (E) Frames (3 μm maximum intensity projection) from a time series showing the example fate of an aPKC-GFP cortical focus (dashed circle) that appeared in the basal cortical hemisphere and dissipated before NEB. Scale bar 2 μm.

DOI: https://doi.org/10.7554/eLife.45815.005

The following source data is available for figure 2:

**Source data 1.** Apical and basal cortical patches.
DOI: https://doi.org/10.7554/eLife.45815.006

hemisphere (defined by the hemisphere opposite where the smaller ganglion mother cell eventually formed; *Figure 2B*). Over time the foci grew into patches, both by fusing with other foci and by the recruitment of additional aPKC from the cytoplasm (*Figure 2A*). Patches generally remained near the location where they initially appeared, undergoing unbiased diffusive movements (*Figure 2A,C, D*). Although cortical targeting by focus formation occurred predominantly in the apical hemisphere,

occasionally we observed foci in the basal hemisphere. However, these foci either dissipated back into the cytoplasm or became part of the apical cap (see below; *Figure 2E*).

## Coordinated flow of cortical aPKC patches leads to formation of a metaphase apical cap

The asymmetric cortical recruitment that occurred in prophase yielded a discontinuous, unorganized apical structure that occupied a large portion of the apical cortical hemisphere. Approximately 90 s before NEB, the aPKC patches on the apical cortex, which had been undergoing uncoordinated, seemingly random movements along the cortex, began to move in a highly coordinated fashion toward the apical pole (*Figures 1B* and *2A,C*; *Figure 1—video 1*). The coordinated movements transformed the broad, discontinuous network of patches into a continuous cap tightly focused around the apical pole. The cap formation process lasted approximately four minutes (3.9 ± 1.1 min; n = 20), measured from the point at which coordinated movement begins to when the continuous apical cap is formed (*Figures 1B* and *2C,D*) with patches traveling a mean distance of 4.1 ± 1.8 μm at a mean velocity of 0.02 ± 0.01 μm/s (n = 12). We term the patch movements 'cortical flow' because they are coordinated, directional, and they occur at the cell periphery, which are characteristics of the motions that take place in the early worm embryo following fertilization when symmetrically cortical aPKC moves toward the anterior cortex (*Munro et al., 2004*; *Wang et al., 2017*). Moreover, as described below, these movements require the actin cytoskeleton. Once the cap is formed it is very stable; we observed little change in aPKC localization over an approximately four-minute period that extended from shortly after nuclear envelope breakdown through metaphase (3.9 ± 0.9 min; n = 20).

## Apical cap disassembly during anaphase causes aPKC spreading to the cleavage furrow

Shortly after the onset of anaphase, the apical cap underwent a dramatic disassembly event that coincided with the changes in cellular morphology that occur at the end of mitosis (*Figure 1B,C*; *Figure 1—video 1*) (*Connell et al., 2011*; *Hickson et al., 2006*). The apical cap, which up until this point had remained uniform, began to break apart into individual patches, similar in appearance to those observed before cap formation (*Figure 1B* and *Figure 3*; *Figure 1—video 1*). Cap disassembly coincided with the extension of the apical cortex that occurs during late anaphase and was characterized by spreading of the patches along the cortex toward the site of cleavage furrow formation, with the overall process lasting 3.9 ± 1.0 min (n = 20) with patches traveling a mean distance of 6.5 ± 3.3 μm at a mean velocity of 0.04 ± 0.03 μm/s (n = 12). The spreading process appeared similar to the cortical flows that occur during cap formation, although in the basal rather than apical direction. Moreover, the cortex of the budding basal daughter cell did not contain any detectable cortical aPKC signal (*Figure 3A*). At the end of telophase, the patches that remained on the cortex of the apical daughter cell (which retains the neuroblast fate) rapidly decreased in intensity until no detectable cortical signal remained, regenerating the cytoplasmic aPKC state present at the start of the neuroblast polarity cycle (*Figure 3B*; *Figure 1—video 1*).

## Apical retention and cortical flows are mediated by the actin cytoskeleton

The dynamic movements of aPKC during the neuroblast polarization and depolarization led us to suspect that the cortical actin cytoskeleton could play important roles in both processes. To investigate whether F-actin participates in the neuroblast polarity cycle, we exposed neuroblasts in various stages of the polarity cycle to the actin depolymerizing drug Latrunculin A (LatA) and imaged the resulting effects on aPKC dynamics.

In neuroblasts treated with LatA during interphase, aPKC appeared in the apical region in early prophase, but in a manner fundamentally different from untreated neuroblasts. In untreated neuroblasts, apical aPKC recruitment occurred primarily via foci appearance and patch growth (*Figure 1*; *Figure 1—video 1*) but foci appearance and patch growth in treated neuroblasts were significantly less frequent (*Figure 4A* and *Figure 4—figure supplement 1*; *Figure 4—video 1*). Furthermore, the foci that did appear often failed to grow into larger patches compared to foci from untreated neuroblasts (*Figure 4—figure supplement 1*). Following NEB, cortical aPKC rapidly spread into the basal

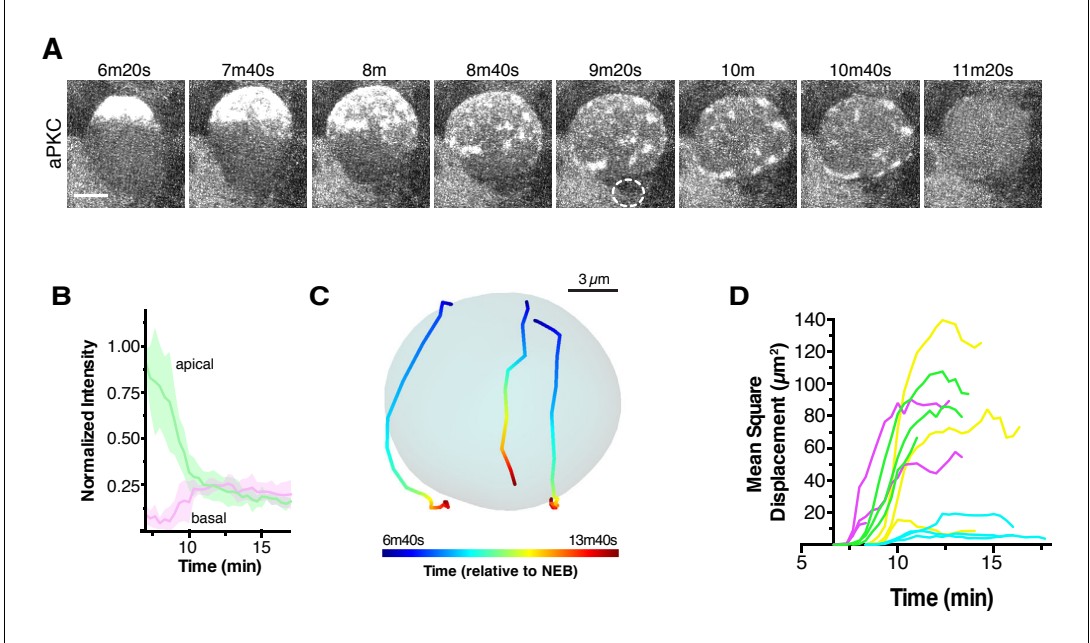

**Figure 3.** Apical cap disassembly. (**A**) Spreading of aPKC-GFP during cap disassembly and patch dissipation. A 6 μm maximum intensity projection (one hemisphere along the apical-basal axis) is shown in each panel. The time is relative to nuclear envelope breakdown. The position of the budding GMC is shown by a dotted circle as identified from the Histone H2A channel (not shown). Scale bar 5 μm. (**B**) Cortical and cytoplasmic intensity of aPKC-GFP in the apical and basal hemispheres during cap disassembly measured from four neuroblasts (error bars represent one standard deviation). Time is shown relative to NEB. (**C**) Example patch trajectories during cap disassembly from particle tracking. Cell outline is shown in light blue. (**D**) Particle tracking of independent patches reveals their mean square displacement as a function of time (relative to nuclear envelope breakdown).
DOI: https://doi.org/10.7554/eLife.45815.007

region before metaphase, a phenomenon that has been previously observed (*Hannaford et al., 2018*), and failed to undergo coalescence.

LatA treatment of prophase neuroblasts allowed us to examine the effect of loss of the actin cytoskeleton when apical aPKC patches are present on the apical cortex. Once treated with LatA, apical patches failed to undergo further growth and prophase treated neuroblasts nearly always failed to undergo coalescence into an apical cap (*Figure 4B* and *Figure 4—figure supplement 1*; *Figure 4—video 2*). Similar to interphase treated neuroblasts, aPKC polarity was lost by cortical spreading into the basal domain before metaphase. Interestingly, cortical patches present at the apical cortex before treatment ceased movement following LatA addition (*Figure 4B,C*; *Figure 4—videos 1* and *2*) and did not spread into the basal domain, indicating that aPKC depolarization results from spreading of non-patch associated protein.

Neuroblasts treated with LatA near the time at which aPKC patches coalesce into an apical cap exhibited limited cap dissociation unlike in untreated neuroblasts where the aPKC cap dissociated into patches that rapidly spread to the cleavage furrow (*Figure 4C* and *Figure 4—figure supplement 1*; *Figure 4—video 3*). In metaphase treated neuroblasts, we observed some breakup of the cap, but most patches remained near the apical pole before dissipating. LatA treated cells also failed to undergo the morphological changes that normally occur during anaphase in which the apical membrane rapidly extends (*Connell et al., 2011*) (*Figure 4C*; *Figure 4—video 3*).

Together, these data indicate that the actin cytoskeleton participates in multiple phases of the neuroblast polarity cycle. First, while the actin cytoskeleton is not required for asymmetric recruitment to the apical cortex, it does play a key role in the discontinuous structure of foci and apical patches that normally form in prophase. Furthermore, the actin cytoskeleton is also required to retain aPKC at the apical cortex as LatA treatment causes aPKC to rapidly spread onto the basal cortex before metaphase, although aPKC that had been incorporated into patches did not appear to migrate into the basal domain (*Figure 4B*; *Figure 4—video 2*). Finally, the rapid dynamics of the

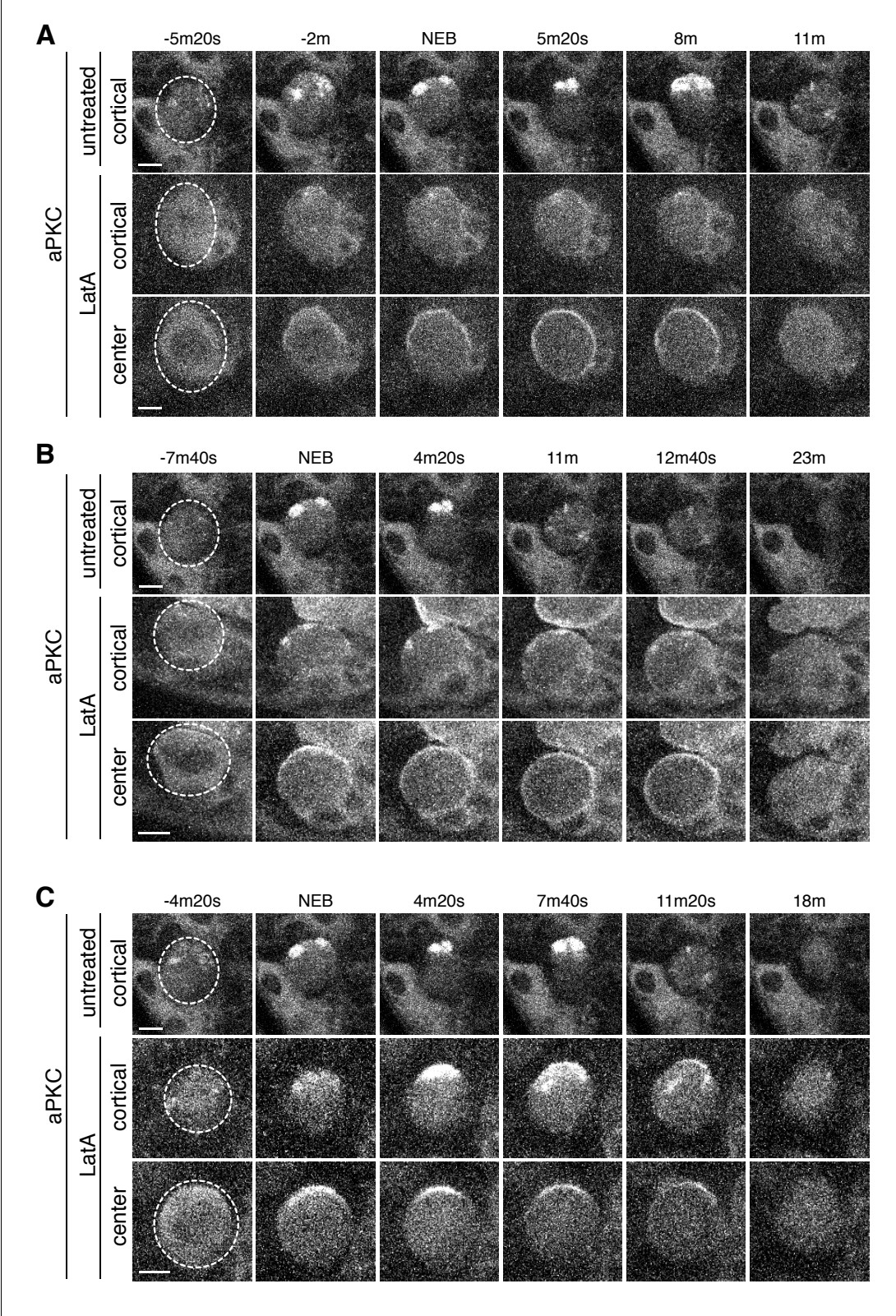

**Figure 4.** aPKC cortical dynamics following disruption of the actin cytoskeleton. (**A**) Effect of treating a neuroblast with LatA beginning in interphase (24m20s prior to NEB) on aPKC localization dynamics. Frames from *Figure 4—video 1* are shown as 4 µm maximum intensity projections along the cortical edge and center of aPKC-GFP taken from *Figure 4—video 1*. The cortical projections from an untreated neuroblast at equivalent time points are shown for reference in the top row. The neuroblast is highlighted by a dashed circle in the first column. Time is shown relative to nuclear envelope

*Figure 4 continued on next page*

*Figure 4 continued*

breakdown (NEB). Scale bar 5 µm. (B) Effect of treating a neuroblast with LatA following the initial cortical recruitment events (7m20s prior to NEB) on aPKC localization dynamics. Frames from *Figure 4—video 2* are shown as in panel A. (C) Effect of treating a neuroblast with LatA following cap coalescence (4 m prior to NEB) on aPKC localization dynamics. Frames from *Figure 4—video 3* are shown as in panel A.

DOI: https://doi.org/10.7554/eLife.45815.008

The following video, source data, and figure supplements are available for figure 4:

**Figure supplement 1.** Quantification of Latrunculin A effects on aPKC localization dynamics.

DOI: https://doi.org/10.7554/eLife.45815.009

**Figure supplement 1—source data 1.** Apical patches in untreated and LatA treated neuroblasts.

DOI: https://doi.org/10.7554/eLife.45815.013

**Figure 4—video 1.** Effect of interphase LatA treatment on aPKC localization dynamics.

DOI: https://doi.org/10.7554/eLife.45815.012

**Figure 4—video 2.** Effect of LatA treatment following cortical recruitment on aPKC localization dynamics.

DOI: https://doi.org/10.7554/eLife.45815.011

**Figure 4—video 3.** Effect of LatA treatment following apical cap coalescence on aPKC localization dynamics.

DOI: https://doi.org/10.7554/eLife.45815.010

apical cap – both its formation via coalescence and its disassembly during anaphase – depend nearly completely on the presence of the actin cytoskeleton.

## Actin-dependent cortical dynamics of the Par complex regulator Bazooka

The polarization of aPKC requires the activity of Bazooka (Baz; aka Par-3) (*Joberty et al., 2000*; *Rolls et al., 2003*; *Tabuse et al., 1998*; *Wodarz et al., 2000*). We analyzed the dynamics of a Baz GFP fusion from a gene trap line (*Buszczak et al., 2007*) to determine if its polarization utilizes similar steps to those we identified for aPKC. We were not able to obtain adequate brightness and photostability with 'red' fluorescent protein variants at the frame rates required to observe aPKC dynamics (except for highly abundant proteins like Histone H2A), precluding simultaneous imaging of both proteins. Imaging of neuroblast asymmetric divisions monitoring Baz-GFP revealed that Baz undergoes dynamics that resemble those of aPKC, but with some noticeable differences (*Figure 5A, B*; *Figure 5—video 1*). Like aPKC, Baz appears to form a discontinuous apical cortical structure during prophase that coalesces to form an apical cap at metaphase with subsequent disassembly. Based on maximum intensity projections of fixed preparations stained with anti-Baz and anti-aPKC antibodies, patches of the two proteins colocalize at early phases of mitosis, although some Baz patches do not have a corresponding aPKC signal (*Figure 5C*).

While Baz's dynamics closely resembled aPKC's, we noticed one significant difference. At mitotic entry aPKC's localization is exclusively cytoplasmic, and while Baz is also found in the cytoplasm during this phase of the cell cycle, we also observed a significant number of cortical puncta (*Figure 5A*; *Figure 5—video 1*). Baz puncta were relatively stationary and many, especially those at the apical cortex, disappeared near mitotic entry. Those with longer lifetimes that persisted into mitosis did not participate in cap coalescence. Shortly after cytokinesis, new puncta often appeared.

We also examined the effect of LatA induced depolymerization of the actin cytoskeleton on Baz's dynamics. In cells treated before metaphase, the appearance of Baz apical patches was reduced following treatment and those that did appear failed to coalesce in most cases, similar to LatA's effect on aPKC dynamics (*Figure 6A,B* and *Figure 6—figure supplement 1*; *Figure 6—videos 1* and *2*). However, while LatA treatment induced spreading of apically enriched aPKC onto the basal cortex, apically recruited Baz remained predominantly in the apical hemisphere following treatment. For neuroblasts treated with LatA near the time of apical cap formation, we observed little Baz cap disassembly, similar to the effect on aPKC's cap (*Figure 6C* and *Figure 6—figure supplement 1*; *Figure 6—video 3*). Our results indicate that the actin cytoskeleton plays a similar role in Baz and aPKC polarization dynamics suggesting that they are polarized by similar mechanisms. However, the actin cytoskeleton appears to be less important for the maintenance of Baz's polarity early in mitosis than it is for aPKC's as LatA did not induce Baz spreading onto the basal cortex.

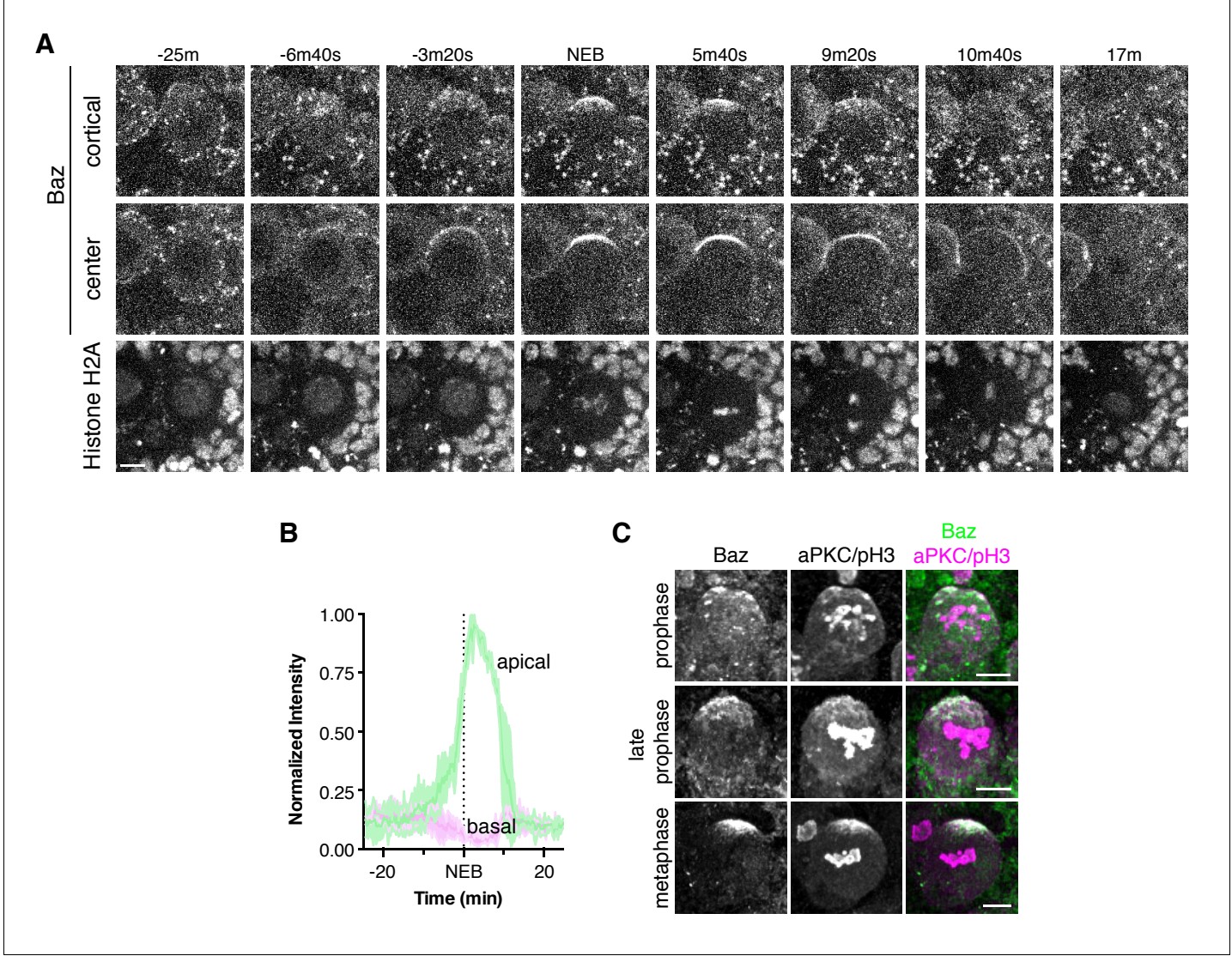

**Figure 5.** Bazooka dynamics during the neuroblast polarity cycle. (**A**) Frames from *Figure 5—video 1* showing 4 µm maximum intensity projections through the cortical edge and center of a larval brain neuroblast expressing Baz-GFP. A central projection of Histone H2A fusion to RFP is shown in the bottom row. The time relative to nuclear envelope breakdown ('NEB') is shown. Scale bar 5 µm. (**B**) Normalized apical and basal cortical intensity (see Materials and methods) of Baz-GFP as a function of time relative to NEB from the divisions of three different neuroblasts with the mean and standard deviation of the signal shown. (**C**) Localization of Baz and aPKC in fixed neuroblasts at early stages of mitosis (pH3 = phospho histone H3).
DOI: https://doi.org/10.7554/eLife.45815.014
The following video is available for figure 5:
**Figure 5—video 1.** Localization dynamics of the Par complex regulator Baz during neuroblast asymmetric division.
DOI: https://doi.org/10.7554/eLife.45815.015

## Discussion

We examined the dynamics that accompany transitions between unpolarized and polarized states of *Drosophila* neuroblasts using rapid imaging throughout the full volume of the cell. These data reveal that canonical neuroblast polarity, with the Par complex's catalytic component aPKC tightly localized around the apical pole at metaphase, results from a multistep process (*Figure 7*). Initially, asymmetric recruitment to the apical cortex leads to a discontinuous structure composed of apical cortical patches. Coordinated cortical flows that begin late in prophase lead to coalescence of the patches into an apical cap. We also discovered a remarkably dynamic depolarization step following metaphase polarity in which the apical cap is broken up into cortical patches that spread to the cleavage

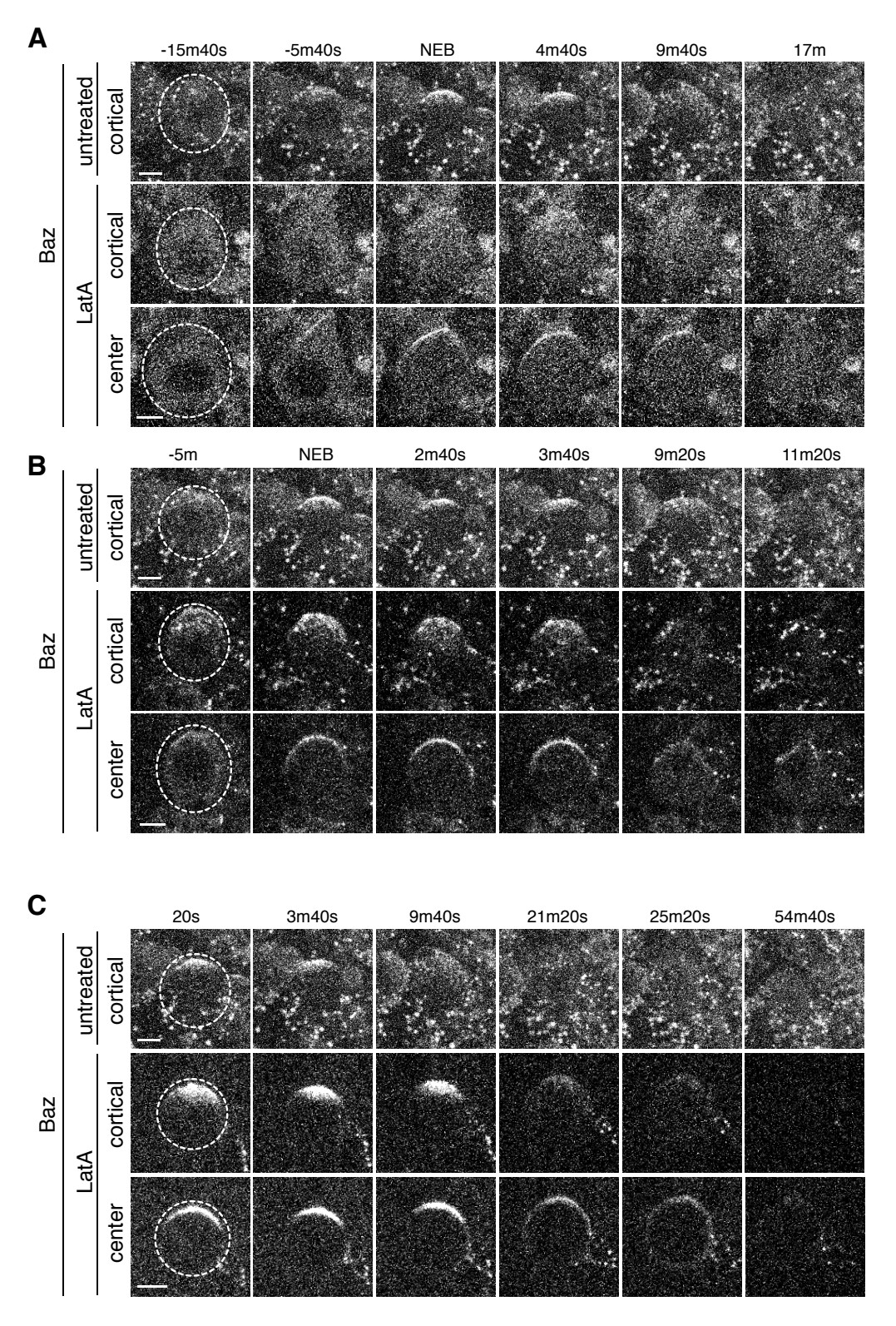

**Figure 6.** Baz cortical dynamics following disruption of the actin cytoskeleton. (**A**) Effect of treating a neuroblast with LatA beginning in interphase (83m20s before NEB) on Baz localization dynamics. Frames from *Figure 6—video 1* are shown as 4 µm maximum intensity projections along the cortical edge and center of Baz-GFP taken from *Figure 4—video 1*. The cortical projections from an untreated neuroblast at equivalent time points are shown for reference in the top row. The neuroblast is highlighted by a dashed circle in the first column. Time is shown relative to nuclear envelope

*Figure 6 continued on next page*

*Figure 6 continued*

breakdown (NEB). Scale bar 5 µm. (**B**) Effect of treating a neuroblast with LatA following the initial cortical recruitment events (7m40s prior to NEB) on Baz localization dynamics. Frames from ***Figure 6—video 2*** are shown as in panel A. (**C**) Effect of treating a neuroblast with LatA following cap coalescence (30 s prior to NEB) on Baz localization dynamics. Frames from ***Figure 6—video 3*** are shown as in panel A.

DOI: https://doi.org/10.7554/eLife.45815.016

The following video and figure supplement are available for figure 6:

**Figure supplement 1.** Quantification of Latrunculin A effects on Baz localization dynamics.

DOI: https://doi.org/10.7554/eLife.45815.017

**Figure 6—video 1.** Effect of interphase LatA treatment on Baz localization dynamics.

DOI: https://doi.org/10.7554/eLife.45815.018

**Figure 6—video 2.** Effect of LatA treatment following cortical recruitment on Baz localization dynamics.

DOI: https://doi.org/10.7554/eLife.45815.019

**Figure 6—video 3.** Effect of LatA treatment following apical cap coalescence on Baz localization dynamics.

DOI: https://doi.org/10.7554/eLife.45815.020

furrow and ultimately dissipate back into the cytoplasm (***Figure 7***). We examined the role of the actin cytoskeleton in the steps that make up the neuroblast polarity cycle and found that it is critical for several different aspects of polarization and depolarization.

In principle, cortical polarity could result from directional cortical flow of initially symmetric cortical molecules, or from asymmetric cortical targeting directly from the cytoplasm. In the early worm embryo, aPKC is initially symmetrically localized to evenly distributed cortical foci (***Munro et al., 2004***; ***Wang et al., 2017***). The cortical cue provided by sperm entry induces anterior directed flows that deplete aPKC foci from the posterior cortex and concentrate it in the anterior hemisphere

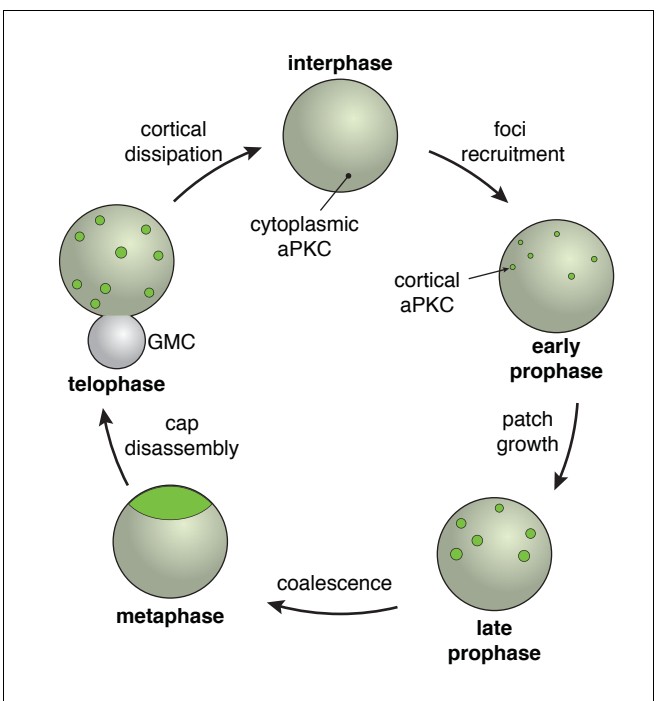

**Figure 7.** The neuroblast polarity cycle. The cycle begins with discontinuous patches of aPKC forming in the apical hemisphere via recruitment to the cortex from the cytoplasm. The aPKC cap observed in metaphase neuroblasts is formed from coordinated cortical flows that lead to coalescence of the discontinuous patches into a uniform structure tightly localized around the apical pole. During anaphase, the cap is disassembled leading to discontinuous spreading that extends to the cleavage furrow, followed by cortical dissipation back into the cytoplasm.

DOI: https://doi.org/10.7554/eLife.45815.021

(**Rose and Gonczy, 2014**). In contrast to the early worm embryo, neuroblasts begin their polarization cycle with cytoplasmic aPKC (**Figure 7**) such that asymmetry in the cortical recruitment process could be sufficient for polarization. We observed that neuroblast polarity begins with asymmetric recruitment but that this process alone leads to a discontinuous polarized structure in the apical hemisphere. Coordinated cortical flows toward the apical pole that begin near metaphase and resemble the polarization of the early worm embryo, transform this unorganized structure into the tightly focused metaphase polarity state. Thus, neuroblast polarity results not from a single process, but from the stepwise activity of two very different cellular processes: asymmetric targeting and actin-dependent cortical flow.

Given that neuroblasts undergo repeated asymmetric divisions, the neuroblast polarity cycle also includes a depolarization step to regenerate cytoplasmic aPKC, the initial state in the cycle. Rather than directly returning to the cytoplasm from the apical cap, we observed a dramatic cap disassembly step that appears similar to the assembly step but in reverse: the cap breaks up into aPKC patches that move toward the basal rather than apical pole. We speculate that cap disassembly may play an especially important role in segregating fate determinants by extending aPKC fully along the cortex to the cleavage furrow, but not beyond. Cortical spreading of aPKC could provide a mechanism for ensuring basal fate determinants such as Miranda and Numb are completely excluded from the cortex that becomes part of the self-renewed neuroblast following cytokinesis. Is cap disassembly an active process? As it is initiated precisely when the dramatic morphologic changes in anaphase occur (**Connell et al., 2011**), it may be that this step utilizes a passive mechanism, in which disassembly is driven by the mechanical stresses that the cortex undergoes during this step of the cell cycle.

The cycle we have identified here represents a new framework for understanding the mechanisms that regulate neuroblast polarity. We have begun to utilize this framework to examine the role the actin cytoskeleton plays in the polarity cycle. While the actin cytoskeleton has been known to be required for metaphase polarity for some time with normally apical proteins such as Inscuteable becoming fully cortical at metaphase when actin filaments are depolymerized (**Broadus and Doe, 1997**), its precise role has been unclear. Here we find that the fully cortical depolarized state can result from a polarized intermediate: in interphase treated neuroblasts aPKC is asymmetrically recruited during prophase but rapidly spreads onto the basal cortex. Thus, at least for aPKC, the actin cytoskeleton is not required for polarized cortical recruitment, but is instead necessary for retention at the apical cortex. Treatment of neuroblasts with LatA at various stages of the cell cycle also revealed that the coalescence of aPKC and Baz patches into a metaphase apical cap and cap disassembly both require an intact actin cytoskeleton. We suspect that the analysis of other perturbations in terms of the neuroblast polarity cycle, such as mutants of previously described polarity genes, will lead to new insight into the mechanisms by which animal cells become polarized.

# Materials and methods

**Key resources table**

| Reagent type (species) or resource | Designation | Source or reference | Identifiers | Additional information |
|---|---|---|---|---|
| Genetic reagent (*Drosophila melanogaster*) | aPKC-GFP | François Schweisguth Lab; **Besson et al., 2015** | | BAC encoded aPKC-GFP |
| Genetic reagent (*D. melanogaster*) | Baz-GFP | Carnegie Protein Trap Library; **Buszczak et al., 2007** | | protein trap line |
| Genetic reagent (*D. melanogaster*) | H2A-RFP | Bloomington Drosophila Stock Center | BDSC:23650; FLYB:FBst0023650; RRID:BDSC_23650 | FlyBase symbol: w[*]; P{w[+mC]=His2A-mRFP1}III.1 |
| Antibody | anti-aPKC (rabbit polyclonal) | Santa Cruz Biotechnology | Santa Cruz: C-20 (SC-216); RRID:AB_2300359 | (1:1000) |

*Continued on next page*

*Continued*

| Reagent type (species) or resource | Designation | Source or reference | Identifiers | Additional information |
|---|---|---|---|---|
| Antibody | anti-Baz (guinea pig polyclonal) | Chris Doe lab; *Siller et al., 2006* | | (1:1000) |
| Antibody | anti-phospho Histone H3 (rabbit polyclonal) | Millipore | Millipore:06–570; RRID:AB_310177 | (1:2000) |
| antibody | Alexa 405- or 647- secondaries | Jackson ImmunoResearch Laboratories | | (1:500) |
| antibody | Alexa 488- secondary | Invitrogen, ThermoFisher Scientific | | (1:500) |

## Fly strains and genetics

*Oregon R* flies were used for examining the localization of fixed endogenous proteins. For live imagine, BAC-encoded aPKC-GFP flies (*Besson et al., 2015*) and a Baz GFP gene trap line (*Buszczak et al., 2007*) were used for assessing aPKC and Baz localization and dynamics, respectively. Each were crossed with a His2A-RFP line (Bloomington stock 23650).

## Live imaging

Third instar larvae were dissected to isolate the brain lobes and ventral nerve cord, which were placed in Schneider's Insect media (SIM). Larval brain explants were placed in lysine-coated 35 mm cover slip dishes (WPI) containing modified minimal hemolymph-like solution (HL3.1). Treated and untreated explants were imaged on a Leica DMi8 microscope (100 × 1.4 NA oil-immersion objective) equipped with a Yokogawa CSU-W1 spinning disk head and dual Andor iXon Ultra camera. Explants expressing aPKC-GFP or Baz-GFP were illuminated with 488 nm and 561 nm laser light throughout 41 optical sections with step size of 0.5 µm and time interval of 20 s. To examine the role of F-actin in aPKC and Baz dynamics, explants were treated with 50 µM LatA (2% DMSO) during imaging.

## Immunofluorescent staining

Intact brain lobes and ventral nerve cord dissected in SIM from third instar *Oregon R* larvae were fixed in 4% PFA and stained with rabbit anti-PKC ζ primary (C20; 1:1000; Santa Cruz Biotechnology Inc) and 647 anti-rabbit secondary antibodies (Jackson ImmunoResearch Laboratories) to determine native aPKC localization. Native Baz localization was assessed in third instar *Oregon R* larval brains that were fixed and stained with guinea pig anti-Baz primary (1:1000; *Siller et al., 2006*) and 488 anti-guinea pig secondary antibodies (Invitrogen). The cell cycle stage was assessed with rabbit anti-phospho Histone H3 primary (1:2000; Millipore) and 405 anti rabbit secondary (Jackson ImmunoResearch Laboratories). Confocal images were acquired on an Olympus Fluoview FV1000 microscope equipped with a 40 × 1.3 NA oil-immersion objective.

## Image processing and visualization

Movies were analyzed in ImageJ (using the FIJI package) and in Imaris (Bitplane). Neuroblasts whose apical-basal polarity axis are positioned parallel to the imaging plane were cropped out to generate representative images and movies. Cortical edge and central maximum intensity projections (MIP) were derived from optical slices capturing the surface and center of the cell, respectively. Optical sections capturing the whole of the cell were assembled for 3D rendering and visualization in Imaris. These volumetric representations were used to quantify the time interval of each process within the polarity cycle and the number of patches recruited to the cortex before cortical flow. Patches that were 0.85 µm$^2$ or larger were selected for quantification.

## Intensity measurements

Intensity profiles were measured in FIJI using a 3 µm line across the apical and basal cortex of 4 µm maximum intensity projections through the center of the neuroblast. Mean signal intensities at time $t$ are normalized using the following equation:

$$I_{normalized}(t) = \frac{I_{mean}(t) - I_{min}}{I_{max} - I_{min}}$$

where $I_{mean}$ is the average intensity within the region specified by the line scan at time $t$, $I_{min}$ is the minimum mean intensity measured across the entire dataset, and $I_{max}$ is the maximum mean intensity measured across the entire dataset.

## Particle tracking

The timing of dynamic events within the aPKC polarity cycle was determined using the H2A channel. Cortical patches were tracked through cap formation and cap dissociation using the Imaris Spots module. Tracking was restricted by an intensity threshold set by the average intensity of the apical cap in metaphase to help increase accuracy of the tracking algorithm. Smaller, lower intensity foci that grew into patches were tracked manually and their tracks were linked to that of the corresponding patches to construct a fully assembled, continuous track. Statistical data such as total patch displacement, mean square patch displacement, and relative speed between time points were collected from the final tracking result. Mean patch speed for cap assembly were calculated using a 160 s time window starting at the onset of cortical flow. Mean patch speed for cap disassembly was calculated using a 160 s time window starting at the onset of disassembly.

## Additional information

### Funding

| Funder | Grant reference number | Author |
| --- | --- | --- |
| National Institute of General Medical Sciences | GM127092 | Ken Prehoda |

The funders had no role in study design, data collection and interpretation, or the decision to submit the work for publication.

### Author contributions

Chet Huan Oon, Conceptualization, Data curation, Formal analysis, Investigation, Visualization, Methodology, Writing—review and editing; Kenneth E Prehoda, Conceptualization, Software, Formal analysis, Supervision, Funding acquisition, Visualization, Methodology, Writing—original draft, Project administration, Writing—review and editing

### Author ORCIDs

Kenneth E Prehoda (iD) https://orcid.org/0000-0003-4214-6158

### Decision letter and Author response

Decision letter https://doi.org/10.7554/eLife.45815.024
Author response https://doi.org/10.7554/eLife.45815.025

## Additional files

### Supplementary files

• Transparent reporting form
DOI: https://doi.org/10.7554/eLife.45815.022

### Data availability

All data generated or analysed during this study are included in the manuscript and supporting files.

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
