## [Decision Letter]

Thank you for submitting your article "Asymmetric recruitment and cortical flow drive the neuroblast polarity cycle" for consideration by *eLife*. Your article has been reviewed by three peer reviewers, including Yukiko M Yamashita as the Reviewing Editor and Reviewer #1, and the evaluation has been overseen by Utpal Banerjee as the Senior Editor. The following individual involved in review of your submission has also agreed to reveal their identity: Hongyan Wang (Reviewer #2).

The reviewers have discussed the reviews with one another and the Reviewing Editor has drafted this decision to help you prepare a revised submission.

In this manuscript, the authors present research that advances our understanding of how *Drosophila* neuroblasts divide asymmetrically by polarizing the Par proteins- aPKC and its upstream regulator Bazooka to the apical cortex during mitosis. Although these proteins have long been known to regulate cell polarization, the mechanisms by which this dynamic process takes place is not well understood. Using live imaging of endogenously expressed and GFP-tagged aPKC during its polarization in dividing *Drosophila* neuroblasts, it is shown to be recruited the apical cortex as clusters that individually accumulate aPKC and also coalesce into larger clusters. For formation of the metaphase apical crescent of aPKC, the clusters move apically in a concerted flow. After metaphase, the condensed apical crescent fragments into clusters that then move basally to the cytokinetic furrow that separates the neuroblast from the ganglion mother cell, and subsequently the clusters dissociate. By applying an inhibitor at distinct cell cycle stages, actin is shown to support aPKC cluster numbers but especially their apical flows and basal dispersions, but actin has a relatively minimal effect on apical cortical recruitment of non-clustered aPKC. Finally, overexpression of another member of the Par complex, Bazooka/Par-3, shows clusters and cluster movements similar to those of aPKC.

This paper is interesting for its close examination of steps of aPKC polarization in an important model of cell polarization. The analysis is well done and support the conclusions. However, there are some outstanding questions and experiments that could make the manuscript much stronger.

Essential revisions:

Overall, the conclusion is interesting and worth reporting *eLife*. However, all the reviewers felt that the manuscript may benefit from fuller analysis. The reviewers noted the lack of obvious experiments, including the following examples. Although all of the suggested experiments do not have to be conducted, the reviewers would like more comprehensive data to be included such that their conclusion is more rigorously supported. The authors can choose which experiments (a few of them) are most critical to address overall concerns raised by reviewers, but the revised version must support the overall conclusion more fully.

1) The authors used BAC-encoded aPKC-GFP from the Schweisguth lab, but in the same paper that reported this stock the Schweisguth lab also reported BAC-encoded Baz-GFP. It is not clear why the current paper didn't report imaging of the BAC-encoded Baz-GFP. If it could not be effectively detected, as the authors state for RFP-tagged versions of Baz, then perhaps there is much less endogenous Baz in the clusters compared to aPKC. If possible, data for the BAC-encoded Baz-GFP should be shown and/or discussed. What about Par-6?

2) Although dual color live imaging for Baz and aPKC may be difficult, it would be important to confirm their relative localization patterns with fixed samples. Especially now that they have better spatial resolution (cortical localization, in addition to 'central' localization), it seems important to confirm Baz and aPKC's localization relationship to support their conclusion.

3) For better understanding of how actin has its effects on aPKC cluster movements, live imaging of actin should be conducted in wild-type neuroblasts at the times when the apical aPKC crescent forms and disperses.

4) F-actin depolymerization via LatA treatment at various stages caused aPKC foci to appear less frequently and prevented the retention of aPKC at the apical membrane. However, the authors find that actin is not responsible for targeting aPKC to the apical membrane. This finding seemed to be inconsistent with an earlier report (Broadus and Doe, 1997) that actin is required for asymmetric localization of Inscuteable, another apical protein. Please provide an explanation for the discrepancy.

5) Is myosin II or microtubules required for the cortical dynamics of the Par complex? Chen K, at al., J. of Cell Biol. 2016 showed that apical aPKC localization in neuroblasts is dependent on MTs. Using a myosin II inhibitor or colchicine treatment for MT depolymerization may provide more information in understanding this process. Also, it would be interesting to see what effect stabilization of F-actin would have on the cortical flow of aPKC.

6) The localization dynamics of Baz resembled those of aPKC. Does F-actin depolymerization influence Baz dynamics similar to aPKC? Furthermore, the authors visualize a different pool of Baz puncta during interphase which do not participate in cap formation. It would be interesting to know what the function of this discrete pool is, possibly by a FRAP experiment.

7) Would be helpful to have a control panel with no LatA treatment in Figure 4.

Revisions to be addressed by textual modifications:

1) The writing needs to be improved. There are many long sentences with ambiguous meaning throughout the manuscript, which made the reading difficult. For example, "However, LatA's effect on the cap disassembly process.… – LatA treated cells did not undergo.….". In addition, I would like to see a sentence of summary at the end of the paragraph).

2) I am somewhat puzzled by aPKC localization on the cortical side and center (Figure 1B) – hard to imagine how they can be continuous. We have been so used to see the clear crescent shape, and now the authors present that there are lots of 'patches' if you just change the focal plane. Not that I think the authors are wrong, but it would help readers in the field if the authors can explain why this was missed before. 3D cartoon representation might help.

3) The persistent interphase Baz puncta might be the result of the over-expression of the Baz-eYFP. A comment about this and other possible caveats of the over-expression should be added to the text.

---

## [Author Response]

Essential revisions:Overall, the conclusion is interesting and worth reporting eLife. However, all the reviewers felt that the manuscript may benefit from fuller analysis. The reviewers noted the lack of obvious experiments, including the following examples. Although all of the suggested experiments do not have to be conducted, the reviewers would like more comprehensive data to be included such that their conclusion is more rigorously supported. The authors can choose which experiments (a few of them) are most critical to address overall concerns raised by reviewers, but the revised version must support the overall conclusion more fully.

We would like to thank the reviewers for their thoughtful comments. One reason we are excited about our findings is that they raise many questions about neuroblast polarity while at the same time providing a new framework for answering them. Along these lines, the reviewers suggested many worthwhile experiments and, as suggested by the editor, we have selected from among them those that would allow us to more fully support the overall conclusions. Our results and the concomitant additions to the manuscript are described in more detail below but the most significant additions are the use of a gene trap line for monitoring Baz localization when expressed from its endogenous promoter (replacing the UAS driven expression data from the original manuscript), and an analysis of LatA treatment on Baz localization. Furthermore, we have extensively edited the text and figures according to the reviewers’ comments and hope that the revised manuscript is more clear and accessible.

1) The authors used BAC-encoded aPKC-GFP from the Schweisguth lab, but in the same paper that reported this stock the Schweisguth lab also reported BAC-encoded Baz-GFP. It is not clear why the current paper didn't report imaging of the BAC-encoded Baz-GFP. If it could not be effectively detected, as the authors state for RFP-tagged versions of Baz, then perhaps there is much less endogenous Baz in the clusters compared to aPKC. If possible, data for the BAC-encoded Baz-GFP should be shown and/or discussed. What about Par-6?

This is an excellent suggestion and while we were not able to obtain the BAC Baz-GFP line from the Schweisguth lab in time for the revision, we were able to use a GFP gene trap line (Buszczak et al., 2007) to observe the localization of Baz expressed from its endogenous promoter. We replaced the UAS-eYFP-Baz data with the data from this line. We noticed no differences in the localization dynamics of Baz between the two lines. We are not aware of a GFP-Par-6 line expressed from its endogenous promoter.

2) Although dual color live imaging for Baz and aPKC may be difficult, it would be important to confirm their relative localization patterns with fixed samples. Especially now that they have better spatial resolution (cortical localization, in addition to 'central' localization), it seems important to confirm Baz and aPKC's localization relationship to support their conclusion.

We examined Baz and aPKC localization in fixed and stained preparations using maximum intensity projections throughout the cell and with neuroblasts at metaphase and earlier stages of mitosis. These data, included in the revised Figure 5, reveal many Baz and aPKC cortical patches that colocalize, but also some Baz patches that do not have a corresponding aPKC patch.

3) F-actin depolymerization via LatA treatment at various stages caused aPKC foci to appear less frequently and prevented the retention of aPKC at the apical membrane. However, the authors find that actin is not responsible for targeting aPKC to the apical membrane. This finding seemed to be inconsistent with an earlier report (Broadus and Doe, 1997) that actin is required for asymmetric localization of Inscuteable, another apical protein. Please provide an explanation for the discrepancy.

The reviewers raise an interesting issue that we have addressed in the Discussion of the revised manuscript. Our analysis of LatA treated neuroblasts reveals that aPKC is initially recruited to the apical cortex but subsequently spreads into the basal domain to become uniformly cortical at metaphase. Broadus and Doe found that in central sections through latrunculin treated metaphase neuroblasts, the apical protein Inscuteable was uniformly cortical 82% of the time (see their Figure 4). Their analysis of fixed cells is consistent with ours from live imaging (assuming Inscuteable and aPKC behave similarly). For example, in our Figure 4A, the 5m20s column shows nearly completely uniformly cortical aPKC (this can be seen even better in Figure 4—video 1 as the cell cycle stage can also be assessed via the H2A signal). We believe the comparison of our results emphasizes the importance of analyzing neuroblast phenotypes by live imaging as it shows that the metaphase phenotype (uniformly cortical) results from depolarization of an earlier polarized state rather than direct recruitment to the entire cortex.

4) The localization dynamics of Baz resembled those of aPKC. Does F-actin depolymerization influence Baz dynamics similar to aPKC? Furthermore, the authors visualize a different pool of Baz puncta during interphase which do not participate in cap formation. It would be interesting to know what the function of this discrete pool is, possibly by a FRAP experiment.

We analyzed the effect of LatA treatment on Baz dynamics and report these results in the revised manuscript (new Figure 6, Figure 6—figure supplement 1, and associated videos). These results are very similar to the effect on aPKC’s dynamics, except that disruption of the actin cytoskeleton had less effect on the maintenance of Baz polarity as it did on aPKC (i.e. LatA caused aPKC to spread into the basal hemisphere whereas Baz didn’t spread following treatment). We agree with the reviewer that it would be useful to know if the interphase cortical puncta that Baz forms are important functionally, but until we know more about their physical nature we will not be able to selectively disrupt them.

5) Would be helpful to have a control panel with no LatA treatment in Figure 4.

This is an excellent suggestion. We have added comparison panels from untreated neuroblasts to both Figure 4 (LatA effect on aPKC dynamics) and the new Figure 6 (LatA effect on Baz dynamics). We believe this addition significantly improves the figure by making it much easier to see the effect of LatA addition.

Revisions to be addressed by textual modifications:1) The writing needs to be improved. There are many long sentences with ambiguous meaning throughout the manuscript, which made the reading difficult. For example, "However, LatA's effect on the cap disassembly process.… – LatA treated cells did not undergo.….". In addition, I would like to see a sentence of summary at the end of the paragraph).

We apologize for any confusion and have extensively revised the text with the goal of making it easier to follow and to remove any ambiguous statements paying particular attention to the LatA Results section (it has been nearly completely rewritten with more explanation added to the Discussion section). We hope that the reviewers find the revision to be clear and accessible.

2) I am somewhat puzzled by aPKC localization on the cortical side and center (Figure 1B) – hard to imagine how they can be continuous. We have been so used to see the clear crescent shape, and now the authors present that there are lots of 'patches' if you just change the focal plane. Not that I think the authors are wrong, but it would help readers in the field if the authors can explain why this was missed before. 3D cartoon representation might help.

We have added a short explanation to the first Results section to address this important issue. We believe the discontinuous nature of the early cortical aPKC signal was not discovered until now for two reasons. First, the discontinuous phase of localization is short (approximately five minutes) so can only be seen when collecting data at high temporal resolution. Second, it is difficult to see discontinuous patches in optical sections through the center of the cell compared to those through the edge of the cell. This is most clearly seen by stepping through Figure 1—video 1 where the two views are side-by-side. For example, at the -300s time point, patches are clearly seen in the cortical projection, but are more difficult to see in the central projection.

3) The persistent interphase Baz puncta might be the result of the over-expression of the Baz-eYFP. A comment about this and other possible caveats of the over-expression should be added to the text.

As noted above, we have replaced the data from flies overexpressing Baz with that from a gene trap line.